# Effects of Nano-Bentonite Polypropylene Nanocomposite Films and Modified Atmosphere Packaging on the Shelf Life of Fresh-Cut Iceberg Lettuce

Zeynab Farahanian [1], Nafiseh Zamindar [1,*], Gulden Goksen [2], Nick Tucker [3,*], Saeed Paidari [1] and Elham Khosravi [1]

1    Department of Food Science and Technology, Isfahan (Khorasgan) Branch, Islamic Azad University, Isfahan 81551-39998, Iran
2    Department of Food Technology, Vocational School of Technical Sciences at Mersin Tarsus Organized Industrial Zone, Tarsus University, Mersin 33100, Turkey
3    School of Engineering, University of Lincoln, Brayford Pool, Lincoln LN6 7TS, UK
*    Correspondence: n.zamindar@khuisf.ac.ir (N.Z.); ntucker@lincoln.ac.uk (N.T.)

**Abstract:** In this study, the shelf life of fresh-cut iceberg lettuce (*Lactuca sativa* L.) was evaluated. Lettuce samples were washed with disinfectant agents and sodium hypochlorite and then soaked in an ascorbic acid solution. Next, samples were stored in packaging films containing three levels (1% and 3% and 0% as a control film) of nano-bentonite particles (NBPs) as a filler in a modified atmosphere for 12 days at 4 °C. Various physicochemical parameters such as color, texture, pH, titratable acidity, dehydration, moisture, dry matter, chlorophyll content, microbial quality, and sensory properties were investigated. Results indicated that nano-packaging had a significant ability to maintain the sensory physicochemical properties of lettuce at the fifth (1% nano-composite film) and ninth (3% nano-composite film) days of storage when compared to the control films. The greatest growths of molds and yeasts were observed in the control films, which demonstrates the effectiveness of the application of bentonite nanoparticle fillers.

**Keywords:** iceberg lettuce; nano-packaging; nano-bentonite; *E. coli*; modified atmosphere packaging

## 1. Introduction

In recent years, iceberg lettuce has become an increasingly popular salad vegetable, particularly in developed countries [1]. Generally, this cultivar has a crisp texture with a desirable aroma, and it is widely consumed [2]. Lettuce contains natural antioxidants such as quercetin, camphorol, luteolin and ascorbic acid, which all play crucial roles in human health [3]. Other important compounds in lettuce are flavonoids and phenolic acids, which are considered to be significant preventives against heart disease and cancer [4]. In addition, the fiber in lettuce helps to regulate the digestive system and reduces the risk of colon cancer [5]. However, cutting and crushing operations during the salad production process expose plant cell tissues to oxygen. This results in an increase in respiration, and the combination of phenolic acid and polyphenol oxidase leads to browning of the lettuce tissue [4]. Cutting, peeling, and crushing can all lead to the release of polyphenol enzymes from the lettuce tissue, which oxidize monophenolic compounds in the presence of oxygen and produce new compounds called quinones, which have red and brown pigments [2]. The degradation of chlorophyll pigments is another reason for undesirable color changes in lettuce leaves [6,7]. Tissue damage can lead to the synthesis of volatile aldehydes, which in high concentrations can be responsible for unpleasant product odor. Factors such as the time of year, preparation methods, packaging, temperature, and storage time can all affect the process of browning and the production of volatile compounds in lettuce [8]. Modified atmosphere packaging (MAP) technology is commonly used to limit the growth of

pathogenic microorganisms in food materials [9]. Applying such new packaging methods can help to maintain the quality and shelf life of fresh-cut fruit and vegetables. MAP technology is currently used to extend the shelf life of fresh-cut lettuce by reducing the browning process through a reduction in the respiration rate [10–13].

The choice of packaging film is very important in retaining the nutritional value of lettuce and is the subject of much research interest. Nanotechnology can improve the capabilities of food packaging, thereby ensuring enhanced food quality and consumer health [14]. For example, clay nanoparticles have been widely used in food packaging owing to their low cost, their ready availability, their effective performance, and their processability [15,16]. These nanoparticles produce films with high resistance to light, heat, and gaseous diffusion. Bentonite is a montmorillonite-rich aluminum phyllosilicate mineral and is available as a commercial nanomaterial [17]. The clay layers can create strong bonds with polymers, resulting in improvements in the mechanical properties of films. These nanoparticles (NPs) can also reduce the absorption of solvents in the packed material and increase chemical resistance through the creation of complex and winding paths in the film structure [18]. The addition of nano-clay fillers to polymers can significantly improve antimicrobial properties [19]. In certain nano-composite films enriched with high concentrations of clay, the NPs inhibit diffusion [15]. The improvement in the oxygen-inhibition properties of nano-packaging has facilitated the uptake of these materials for the preservation of fresh foods such as fruits, salads, and ready meals (Figure 1) [18].

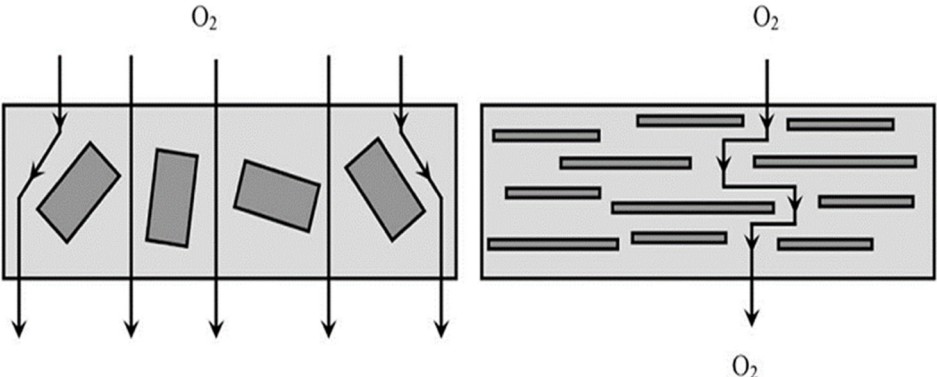

**Figure 1.** Nano-bentonite barrier properties.

The aim of the present study is to evaluate the physicochemical and antimicrobial characteristics of polypropylene films containing various percentages of nano-bentonite particles (NBPs) for application in the modified atmospheric packaging of fresh fruit and vegetables, particularly fresh-cut lettuce samples.

## 2. Materials and Methods

### 2.1. Materials

All chemicals and reagents were of analytical grade. Sodium hydroxide was purchased from Merck (Darmstadt, Germany).

### 2.2. Plant Materials and Treatment

Lettuce (*Lactuca sativa* L.) was purchased from a commercial cultivator in Isfahan, Iran and refrigerated (at 4 °C) for one hour. Then, the damaged leaves, head, and core of the lettuce were removed. Mature leaves were cut into 9–20 mm pieces with a stainless-steel kitchen knife. Then, the fresh-cut lettuce was washed and disinfected using sodium hypochlorite (100 mg/L) and L-ascorbic acid solutions (0.5% $w/w$) at 4 °C. Finally, the excess water was removed from the disinfected samples by performing manual centrifugation (600 rpm, 30 s, Isfahan, Iran).

### 2.3. Preparation of Nano-Packaging

After drying, 100 g of fresh-cut lettuce was packed in nano-packaging light polypropylene with varying percentages of nano-bentonite (0, 1 and 3 wt.%) (Merck, Germany). To equilibrate pack atmospheres, the controlled atmosphere was injected into the packages using a commercial modified atmosphere packaging (MAP) device, and the packaging was sealed [20]. The prepared bentonite nanofilm packaged samples were then transferred to the refrigerator at 4 °C to evaluate their physicochemical properties over 12 days of storage (0, 3, 5, 9 and 12 days). Three sample replications were made. The containers weighed 10.45 g, with internal dimensions of $11 \times 14.5 \times 5$ cm$^3$. Polypropylene film and polypropylene film containing 3 and 1% (*w/w*) nano-bentonite with a thickness of 580 μm was supplied by Firooz Baspar Yaran Packaging Co. Ltd., (Isfahan, Iran), and used as supplied.

### 2.4. Physicochemical Analyses during Storage

#### 2.4.1. pH Measurement

The pH value was measured using a pH meter (CP501, Elmetron, Zabrze, Poland). For this purpose, a 10 g aliquot of the lettuce sample was completely homogenized and mixed with 50 mL of distilled water. It was then filtered and the solution was used for pH determination [21].

#### 2.4.2. Titratable Acidity (TA)

The determination of titratable acidity was carried out using sodium hydroxide (0.1 mol/L NaOH). In this method, a 10 g aliquot of the packed lettuce sample was homogenized in distilled water (1:1 *w/v*) and centrifuged. The supernatant was filtered, and the titration with 0.1 mol/L NaOH was continued to the end point at pH = 8.2 (Equation (1)) [8,22–24].

$$TA = \%Acid = \frac{(V_{NaOH}) \times (0.1\,N) \times (meq)}{Sample\,(g)} \times 100 \tag{1}$$

where V is the consumption volume of NaOH, N is the normality of NaOH, and meq is milliequivalents of citric acid, respectively.

#### 2.4.3. Exudate Measurement

In this test, a specific part of a lettuce leaf was weighed (W1) and then placed between two filter papers (MN 615, qualitative grade) under a pressure of 11 kg for 10 s. After the application of pressure, the sample was weighed again (W2) [25]. The percentage of leakage was determined by using Equation (2):

$$Exudates\,\% = \frac{W1\,(g) - W2\,(g)}{W2\,(g)} \times 100 \tag{2}$$

#### 2.4.4. Dry Matter

The dry-matter content of samples was measured according to the method reported by Martin-Diana et al. [25]. A piece of fresh-cut lettuce was weighed and transferred into an oven at 100 °C. After 2 h, the sample was weighed again. The percentage of dry matter of the lettuce was evaluated by using Equation (3):

$$Dry\,matter\,\% = \frac{W1\,(g)}{W2\,(g)} \times 100 \tag{3}$$

where W1 and W2 are the final and initial weights of the plates, respectively.

### 2.4.5. Moisture Content (MC)

The moisture content of the lettuce was determined according to (WHO). In this method, the difference between the primary (W1) and the secondary (W2) weight of sample was expressed as the percentage of moisture, as shown in Equation (4):

$$\text{MC \%} = \frac{(W1 - W2)\ (g)}{W1\ (g)} \times 100 \qquad (4)$$

### 2.4.6. Firmness Analysis

The texture quality of the lettuce was evaluated using a histometer (SanTam STM-1 ENG.Design co. Ltd). The lettuce samples were subjected to the pressure of a probe with a diameter of 6 mm and a speed of 60 mm/min. The maximum load for each sample was recorded on its peak point (kN), which expressed the point of rupture of the lettuce sample [25].

### 2.4.7. Color Assessments

The evaluation of the color properties of a fresh-cut lettuce on each day of the experiment was performed using a colorimeter (Hunter Lab CR-400 color difference meter, Konica Minolta, Tokyo, Japan) according to Ghorbani et al. [26]. The color parameters L (brightness), a* (red-green), and b* (yellow-blue) were measured. The degree of color changes ($\Delta E$) on different days was expressed by means of comparison to measurements taken on the first day of packaging (Equation (5)) [27].

$$\Delta E = [(\Delta L)^2 + (\Delta a^*)^2 + (\Delta b^*)^2]^{1/2} \qquad (5)$$

### 2.4.8. Determination of Chlorophyll Concentration

To investigate the concentration of chlorophyll, 1 g of lettuce aliquot was mixed thoroughly with 10 mL of extraction solution (80% acetone-water) in a centrifuge tube. The resulting suspension was then centrifuged for 10 min at 6000 rpm. The collected supernatant was mixed with more extraction solution to extract any remaining pigment by means of repeating the above procedure. Finally, the clear solution was diluted with 80% acetone in separate 25 mL volumetric flasks. The chlorophyll concentration was determined by performing spectrophotometry (S2100 Unico) at wavelengths of 645 and 663 nm, and the values of chlorophyll a, b and the total were measured using Equations (6)–(8) [28]:

$$\text{Mg chlorophull a} = (12.7 \times A_{663} - 2.69 \times A_{645})\ V/1000\ W \qquad (6)$$

$$\text{Mg chlorophull b} = (22.9 \times A_{645} - 4.68 \times A_{663})\ V/1000\ W \qquad (7)$$

$$\text{Mg chlorophull a} = (20.2 \times A_{645} - 8.02 \times A_{663})\ V/1000\ W \qquad (8)$$

where V is volume of supernatant, A is absorption in wavelengths 663, 645 and 470 nm, and W is the weight of sample (g).

### 2.4.9. Sensory Analysis

The sensory analysis of the lettuce during storage included an evaluation of the freshness of appearance, browning, off-flavors, and the texture of the leaves; this was carried out by a trained panel of 10 evaluators aged from 25 to 40 years. Scoring was carried out by ranking the performance of the lettuce on a scale of 1–5 for browning, tenderness, taste and odor; in scoring conditions, ideal freshness was ranked 1, acceptable freshness was ranked 3, and poor freshness was ranked 5. In scoring the appearance of the lettuce, a 1–9 scale was used where the 1st ranking was considered excellent, the 5th ranking was considered relatively good, and the 9th ranking was considered poor [21].

*2.5. Microbial Analysis*

2.5.1. Mold and Yeast Count Test

Samples were periodically taken to quantify mold and yeast contamination using the method proposed by Fukumoto et al. [29]. A 1 g aliquot of each sample was transferred to a test tube and diluted ($10^{-2}$, $10^{-4}$, $10^{-6}$, and $10^{-8}$) with 0.9% (*w/v*) NaOH solution. After the complete homogenizing of the samples, they were transferred to plates containing a sterile culture medium of dichloran rose-bengal chloramphenicol agar (DRBC) and incubated (DIAP Instrument Type IC55) at 30 °C for 10 days. After this period, the mold and the yeast were counted. Each treatment was performed with three replications [30].

2.5.2. *E. coli* Inoculum on Lettuce

The growth rate of the serotype *E. coli* O157: H7 on lettuce during storage in nano-packaging was evaluated according to Anderson et al. [29]. In this method, the *E. coli* strain was purchased from Daroush Company, and a linear culture in Triptic Soy Broth (TSB) medium was carried out to ensure the *E. coli* strain's purity. Then, the bacterium was cultured on eosin methylene blue (EMB) agar, and the purity of the medium was confirmed by evaluating its turbidity using a spectrophotometer (wavelength 625 nm). The lettuce leaf aliquots were sterilized by immersion in sodium hypochlorite solution. The pieces of sterilized leaves ($5 \times 5$ cm$^2$) were then immersed in the microbial solution for 2 min and placed between two pieces of nano-film ($5 \times 5$ cm$^2$). The samples were heat-sealed and stored at 4 °C for 0, 2, 4, 6, 8 and 10 days to further investigate their microbiological characteristics.

2.5.3. *E. coli* Counts

In order to measure the *E. coli* count, suspension was that four different dilutions ($10^{-2}$, $10^{-4}$, $10^{-6}$, and $10^{-8}$) of each sample were prepared in test tubes using 0.9% (*w/v*) peptone water solution. The homogenized samples were cultivated in the culture medium of eosin methylene blue agar (EMB) and incubated at 37 °C for 2 days. Finally, the number of colonies was read from the plate (log CFU/g) [29].

*2.6. Statistical Analysis*

In this study, to evaluate the effect of variables (the level of nano-bentonite and time) on the physicochemical and microbial properties of lettuce in nano-packaging, analysis of variance (ANOVA) was applied. SAS software 18.1 and mean comparisons with the Duncan test and the Microsoft EXCEL spreadsheet were used to perform analysis of variance and to graph the results.

**3. Results**

*3.1. Physicochemical Properties*

The changes in visual appearance of lettuce packed with and without nano-packaging during storage time are shown in Figure 2.

Figure 3A shows the significant effect of time (12 days of storage) on pH ($p = 1\%$). As can be seen, pH values in each packaging variant decreased over the test period. The pH of the packages did not change until the third day of storage. Additionally, there was no significant difference between the pH values of the 3% nano-packaging on the ninth day of storage and on the fifth day. This indicates better maintenance of pH in the 3% nano-packaging compared to the other two packages. In contrast, the lowest pH was seen in the 1% nano-packaging on the twelfth day of storage, while there was no significant difference between the pH values in the control packaging and the 3% nano-packaging. Therefore, these films were both suitable for storing lettuce for 12 days.

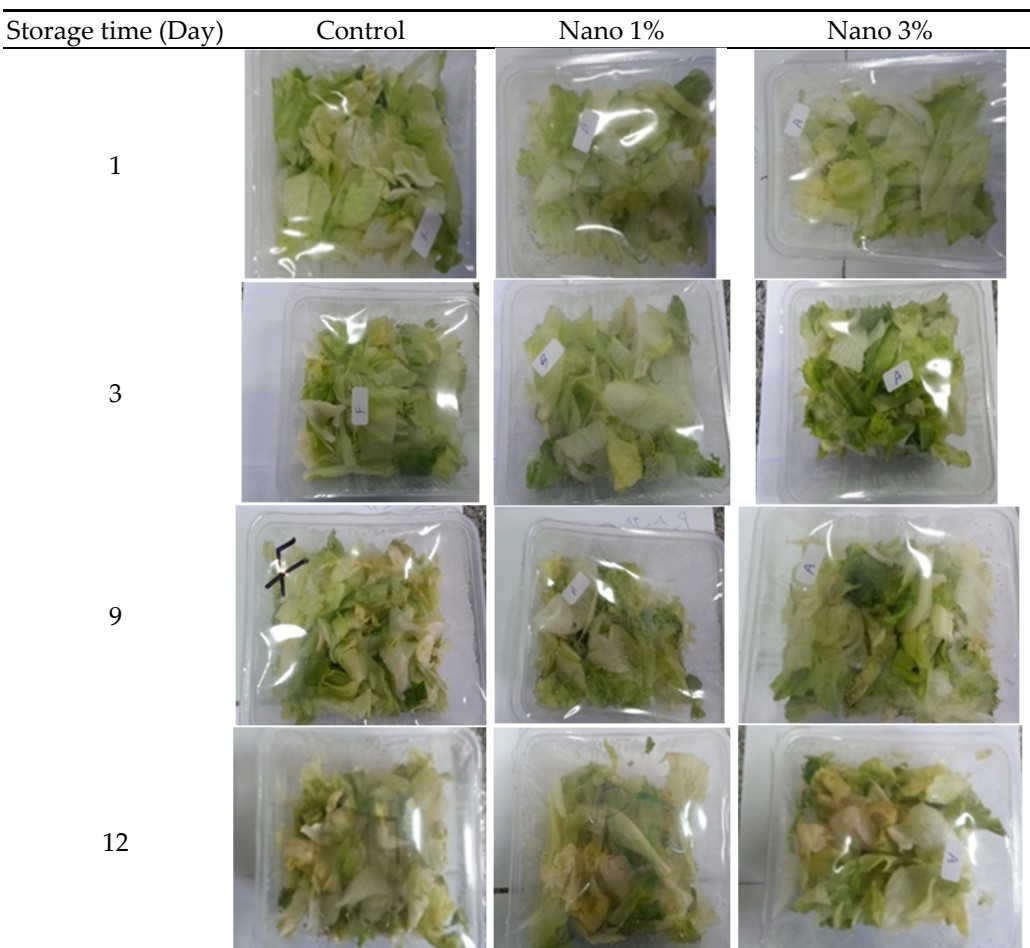

**Figure 2.** Visual appearance of packed lettuce with and without nano-packaging during storage time.

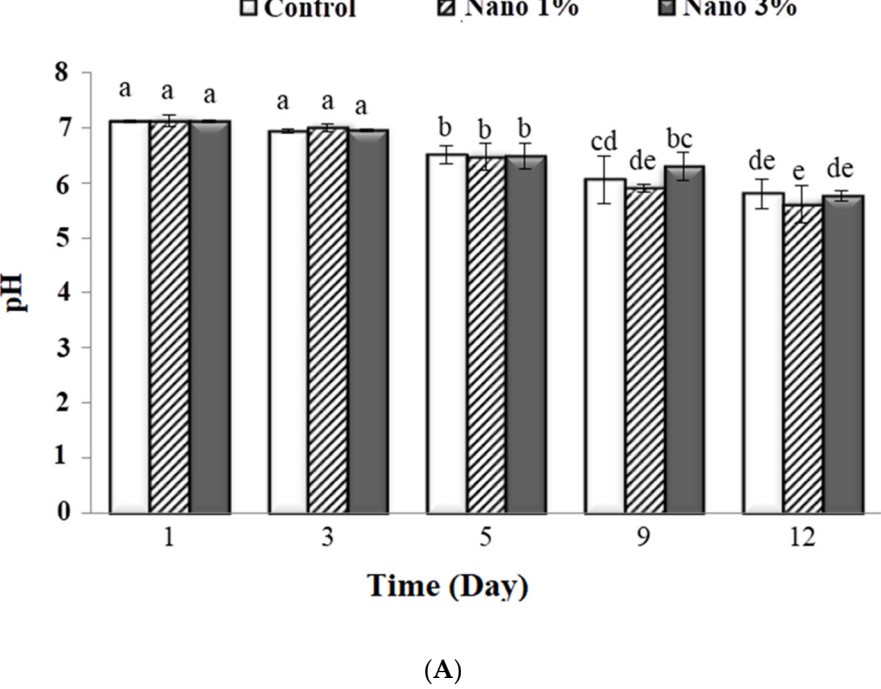

(**A**)

**Figure 3.** *Cont.*

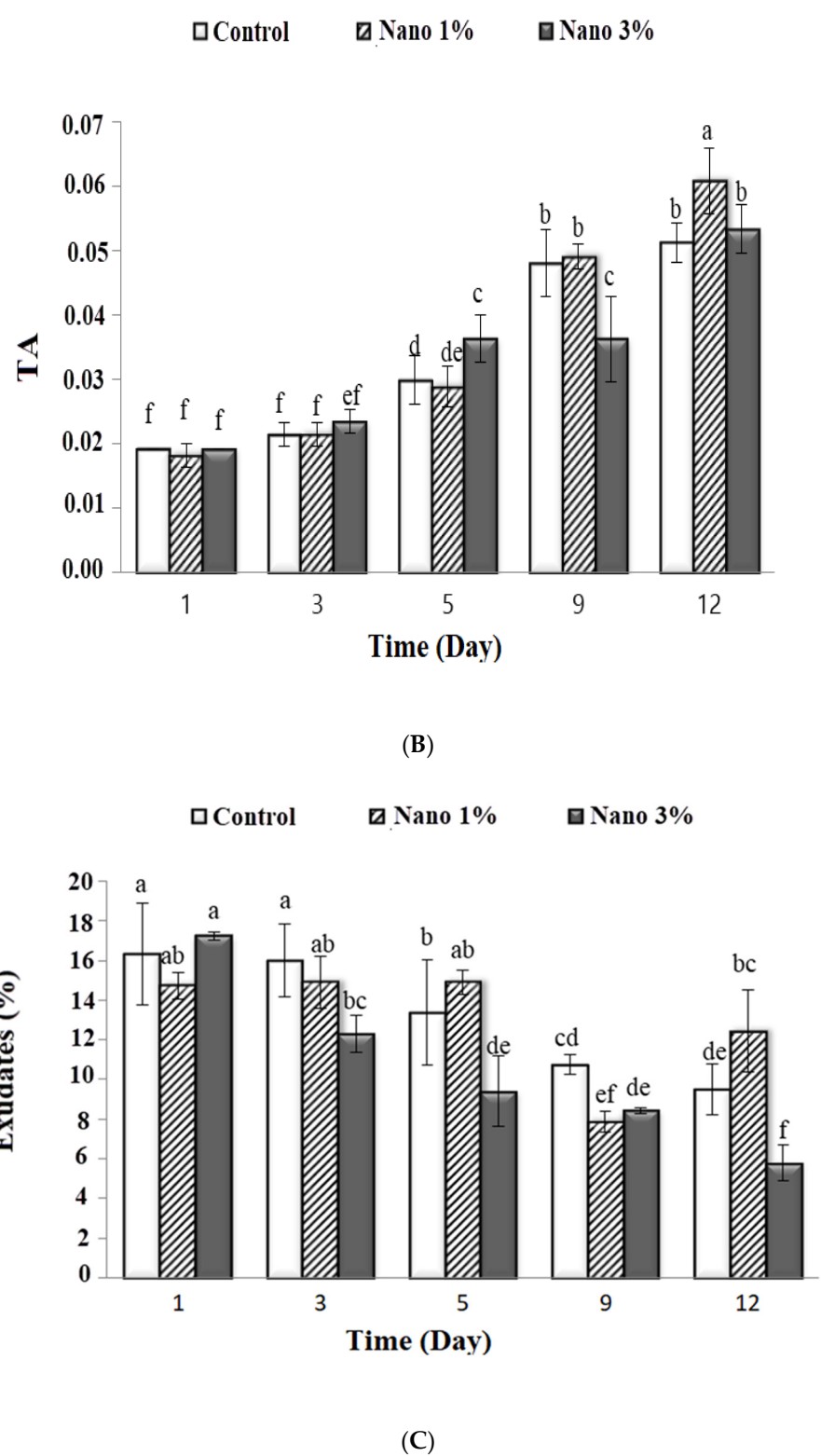

**Figure 3.** (**A**). The effect of storage time on the pH of lettuce in MAP. (**B**). The interaction of packaging × storage time on TA (titratable acidity) of lettuce in MAP. (**C**). The interaction of packaging × storage time on exudates of lettuce in MAP. Different lowercase letters within the same storage time of each treatment denote statistical significant difference.

The results of the interaction between the types of packaging and the storage times, and their effect on titratable acidity (TA), are shown in Figure 3B. The amount of TA in all three types of packaging was stable until the third day of storage. However, on

the fifth day of storage, the amount of TA in the control packaging and the 1% nano-packaging decreased slightly. On the ninth day of storage, the 3% nano-packaging showed better performance in terms of acidity changes when compared to the other two types of packaging. By the twelfth day of storage, the amount of acidity produced in the control nano-packaging and the 3% nano-packaging was the same, and lower than the 1% nano-packaging, indicating that the use of MAP had positive effects on maintaining the quality of the lettuce. There is significant variation in results reported elsewhere. For example, Yang et al. [31] reported that strawberries developed alkaline tendencies during the storage period, whilst lettuce became slightly acidic. These differences may be due to the initial acidic content of strawberries and the neutral content of lettuce. From our results, it can be concluded that there was an inverse relationship between TA and the pH content of the lettuce during storage.

The values of exudate in the packaged lettuce are shown in Figure 3C. The values of exudate in the 1% nano-packaging remained stable between the first and fifth days of storage ($p < 0.05$). Additionally, the highest and lowest values of exudate were obtained in the 1% nano-packaging and the 3% nano-packaging, respectively, on the twelfth day of storage. Therefore, the 3% nano-packaging was suitable for both short (3–5 days) and long-term (9–12 days) storage of lettuce. The results of exudate during 12 days of storage were in accordance with those reported by Martin-Diana et al. [25].

The trend of changes in the percentage of moisture and the dry-matter content of lettuce is shown in Table 1. These results indicate that the control packaging was not suitable for 12 days of storage. However, this packaging showed acceptable results in maintaining the quality of lettuce up to the ninth day of storage. It is worth noting that the nano-packaging had a higher ability to maintain the dry matter of samples compared to the control packaging. Moreover, all three types of packaging revealed similar results in maintaining moisture, whilst on the twelfth day of storage, nano-packaging confirmed its better quality. These results had a linear relationship with the crispness of texture of products. Sugar consumption resulting from the normal metabolic activities of living lettuce tissues may be the reason for the decrease in dry matter during the storage period, but the accumulation of more carbon dioxide in the nanofilms may reduce the metabolic activities of fresh-cut lettuce causing less dry-matter loss and more moisture maintenance [26].

**Table 1.** The interactions of nano-packaging and storage time on dry matter, moisture and texture of lettuce in MAP.

| Nano (%) | Time (Day) | Dry Matter (%) | Moisture (%) | Texture (-) |
|---|---|---|---|---|
| control | 1 | 4.93 ± 0.50 [cd] | 95.07 ± 1.29 [a] | 0.0026 ± 0.0002 [fgh] |
|  | 3 | 4.43 ± 0.35 [d] | 95.57 ± 0.91 [a] | 0.0028 ± 0.0003 [dg] |
|  | 5 | 4.43 ± 0.23 [d] | 95.57 ± 1.21 [a] | 0.0033 ± 0.000 [cd] |
|  | 9 | 5.27 ± 0.15 [bcd] | 94.40 ± 0.70 [a] | 0.0027 ± 0.0003 [eh] |
|  | 12 | 7.47 ± 0.31 [a] | 92.53 ± 0.70 [b] | 0.0025 ± 0.0003 [gh] |
| 1 | 1 | 4.53 ± 0.61 [cd] | 95.47 ± 0.61 [a] | 0.0038 ± 0.0003 [bc] |
|  | 3 | 4.57 ± 0.12 [cd] | 95.43 ± 1.01 [a] | 0.0060 ± 0.0002 [a] |
|  | 5 | 4.30 ± 0.10 [d] | 95.70 ± 1.10 [a] | 0.0032 ± 0.0002 [de] |
|  | 9 | 5.50 ± 1.14 [bc] | 94.50 ± 1.61 [a] | 0.0016 ± 0.0001 [i] |
|  | 12 | 5.00 ± 0.80 [cd] | 95.00 ± 1.80 [a] | 0.0030 ± 0.0003 [def] |
| 3 | 1 | 4.77 ± 0.33 [cd] | 95.32 ± 0.31 [a] | 0.0042 ± 0.0004 [b] |
|  | 3 | 4.40 ± 0.26 [d] | 95.60 ± 0.26 [a] | 0.0025 ± 0.0005 [gh] |
|  | 5 | 4.57 ± 0.67 [cd] | 95.43 ± 0.67 [a] | 0.0030 ± 0.0002 [def] |
|  | 9 | 5.92 ± 0.64 [b] | 94.07 ± 1.18 [ab] | 0.0023 ± 0.0002 [h] |
|  | 12 | 4.67 ± 0.12 [cd] | 95.33 ± 1.10 [a] | 0.0017 ± 0.0002 [i] |

Means with different letters within a column are significantly different ($p < 0.05$).

### 3.2. Texture Parameter

The results of the texture changes in the lettuce during 12 days of storage are shown in Table 1. The measurement of the crispiness of samples was performed using a puncture cell attachment, which applied a force on the texture of the lettuce. Generally, the lower the force required to break the tissue, the softer the tissue [32]. The results of this study indicate that there was no significant difference in the texture parameter between samples up to the fifth day of storage. However, the crispiness of the tissue decreased from the fifth day onwards, and by the ninth day these changes were significant ($p < 0.05$). The main reason for this issue relates to the loss of moisture and the increase in elasticity during storage. The 1% nano-packaging shows a high ability to maintain the quality of the lettuce's texture until the fifth day of storage [33].

### 3.3. Color Parameters

In this study, the interaction (packaging × storage time) on color parameters (a*, b* and L) was investigated, and the results are shown in Table 2. The results of the time effect on the value of brightness showed that the highest brightness was seen on the first day and the lowest brightness value of the lettuce was seen on the twelfth day of storage ($p < 0.05$). A decrease in factor L could be due to the browning process that occurs during storage. Altunkaya et al. reported that in film packages containing whey and control protein, there was no significant difference between the brightness of the whey and the control film packaging [32]. The brightness of the lettuce in all the packaging remained stable until the fifth day, whilst in the 3% nano-packaging no significant difference in the brightness of the leaves was observed until the ninth day of storage. However, it was observed that all three packaging types had the same effect on the lightness retention of the lettuce leaves until the twelfth day of storage.

**Table 2.** The interaction effects of nano-packaging and storage time on the color parameters of lettuce in MAP.

| Nano (%) | Time (Day) | L | a* | b* | ΔE |
|---|---|---|---|---|---|
| control | 1 | 85.62 ± 2.73 [a] | 6.97 ± 0.66 [cd] | 23.12 ± 1.98 [de] | |
| | 3 | 84.73 ± 1.21 | 8.41 ± 1.13 [de] | 33.33 ± 1.56 [cde] | 2.41 ± 0.43 [g] |
| | 5 | 80.43 ± 2.66 [ad] | 8.12 ± 1.25 [de] | 23.98 ± 3.22 [cde] | 6.59 ± 1.29 [e] |
| | 9 | 75.59 ± 0.65 [d] | 9.32 ± 1.10 [ef] | 27.22 ± 1.46 [bcd] | 11.38 ± 1.56 [d] |
| | 12 | 59.70 ± 8.29 [e] | 11.06 ± 1.32 [fg] | 28.87 ± 2.76 [abc] | 27.14 ± 2.19 [b] |
| 1 | 1 | 87.01 ± 0.96 [a] | 6.11 ± 0.61 [bc] | 13.97 ± 1.54 [f] | – |
| | 3 | 87.33 ± 1.70 [a] | 6.77 ± 0.99 [cd] | 17.08 ± 3.16 [f] | 3.35 ± 0.47 [fg] |
| | 5 | 81.62 ± 3.42 [ad] | 9.21 ± 0.49 [ef] | 25.07 ± 4.46 [bcd] | 12.87 ± 1.20 [d] |
| | 9 | 76.90 ± 3.03 [cd] | 12.45 ± 0.72 [gh] | 30.03 ± 1.99 [ab] | 20.31 ± 0.87 [c] |
| | 12 | 60.17 ± 6.87 [e] | 16.17 ± 0.73 [i] | 33.38 ± 0.48 [a] | 34.80 ± 3.41 [a] |
| 3 | 1 | 85.88 ± 2.99 [a] | 4.16 ± 0.20 [a] | 14.44 ± 4.58 [f] | – |
| | 3 | 83.72 ± 3.06 [abc] | 4.40 ± 0.53 [ab] | 18.77 ± 2.98 [ef] | 5.27 ± 0.57 [ef] |
| | 5 | 80.15 ± 4.15 [ad] | 8.10 ± 1.61 [de] | 24.58 ± 0.86 [bcd] | 12.61 ± 1.68 [d] |
| | 9 | 77.45 ± 1.91 [bcd] | 10.47 ± 1.77 [fg] | 30.13 ± 2.50 [ab] | 19.67 ± 1.56 [c] |
| | 12 | 56.69 ± 6.02 [e] | 14.15 ± 1.24 [h] | 29.22 ± 3.12 [ab] | 32.40 ± 1.23 [a] |

Means with different letters within a column are significantly different ($p < 0.05$).

Another color parameter is a*, which indicates a change in color from green to red. The main reason for the decrease in the value of a* is the formation of brown pigments due to enzymatic activity [34]. Overall, there is a linear relationship between chlorophyll content and the browning process. The values in the 1% and the 3% nano-packaging showed a downward trend over the whole period, while in the control packaging there were no significant differences between the third and the twelfth days of storage ($p < 0.05$). Chemical changes resulted in a* in the 1% nano-packaging being allocated the highest value, which

was, however, similar to that of the control packaging. The 3% nano-packaging revealed a notable decrease in the green color of lettuce leaves on the ninth day of storage compared to the fifth day (Table 2). Thus, the nano-packages had the capability to preserve the green color in lettuce leaves until the fifth day of storage.

The result of the b* value in Table 2 indicated that its value in all three packages showed an upward trend between the first day and the twelfth day of storage. The lowest value of b* appeared in the 1% and the 3% nano-packaging on the first day, while the highest value of b* was for the 1% nano-packaging on the twelfth day of storage ($p < 0.05$). The brightness index and the yellowness index in all three packages showed the same levels until the fifth day. After the fifth day, both nano-packets acted almost identically.

Our results indicate that the interaction effects (packing × storage time) on the ΔE values of lettuce were significant at the 1% level. In general, the trend for ΔE value was upward over the given period. However, the rate of these changes was higher in the 1% nano-packaging compared to the others (Table 2). The highest value of ΔE was obtained in the 1% nano-packaging and the lowest in the 3% nano-packaging on the twelfth day, and in the control packaging on the first day of storage. These results were in line with An et al. [35] who studied the effect of nano-packaging on the quality of green asparagus during storage.

### 3.4. Chlorophyll Measurement

The results of the analysis of variance indicated the significance of the interactions (packaging × storage time) on the value of chlorophyll a, chlorophyll b, and the total chlorophyll ($p < 0.05$) (Table 3). Generally, the level of chlorophyll in vegetables during storage varies owing to certain environmental factors such as temperature, light, humidity, oxygen, and atmospheric ethylene levels [36]. Additionally, enzymes such as chlorophyllase and magnesium decalactase play a significant role in reducing the level of chlorophyll pigments during vegetable storage [8,37]. However, reducing the amount of water in plant leaves can lead to increased chlorophyllase enzyme and a consequent decrease in the levels of chlorophyll [38]. From our results, the highest level of total chlorophyll was found to be in the 1% nano-packaging on the first and third days of storage. However, there were no significant changes in the level of chlorophyll on the fifth and ninth days of storage. On the twelfth day, a decrease in the level of total chlorophyll in the control packaging was observed, whilst this level for the two other packages (the 1% and the 3% nano-packaging) did not show any changes. Additionally, the results showed that the 1% and the 3% nano-packaging had a high protective ability to retain chlorophyll in lettuce leaves until the twelfth day of storage compared to the control packaging, which indicated an improvement in the shelf life and in the preservation of chlorophyll in nano-packages. The level of chlorophyll b in the control packaging and the 1% nano-packaging remained stable, but by contrast, there was a significant difference in residual chlorophyll b in the 3% nano-packaging. This means that the control packaging and the 1% nano-packaging have a high ability to preserve chlorophyll b in lettuce leaves. In green asparagus packaging, there were no changes in the level of chlorophyll in the control packaging, whilst in nano-packaging the trend was downward [35].

**Table 3.** The interaction effects of nano-packaging and storage time on the content of lettuce chlorophyll in MAP.

| Nano (%) | Time (Day) | Chlorophyll a | Chlorophyll b | Total Chlorophyll |
|---|---|---|---|---|
| | | (mg/g) | | |
| control | 1 | 0.025 ± 0.005 [de] | 0.012 ± 0.003 [cd] | 0.038 ± 0.008 [c] |
| | 3 | 0.028 ± 0.003 [cd] | 0.015 ± 0.002 [bc] | 0.042 ± 0.002 [bc] |
| | 5 | 0.031 ± 0.000 [cd] | 0.009 ± 0.000 [dg] | 0.039 ± 0.000 [bc] |
| | 9 | 0.007 ± 0.001 [g] | 0.0021 ± 0.002 [a] | 0.029 ± 0.003 [d] |
| | 12 | 0.015 ± 0.000 [f] | 0.003 ± 0.001 [h] | 0.019 ± 0.001 [e] |
| 1 | 1 | 0.034 ± 0.007 [bc] | 0.024 ± 0.006 [a] | 0.058 ± 0.012 [a] |
| | 3 | 0.042 ± 0.004 [a] | 0.017 ± 0.002 [b] | 0.059 ± 0.006 [a] |
| | 5 | 0.030 ± 0.000 [cd] | 0.009 ± 0.000 [de] | 0.040 ± 0.000 [bc] |
| | 9 | 0.019 ± 0.001 [f] | 0.005 ± 0.001 [fgh] | 0.024 ± 0.000 [de] |
| | 12 | 0.017± 0.003 [f] | 0.004 ± 0.001 [h] | 0.020 ± 0.000 [de] |
| 3 | 1 | 0.030 ± 0.008 [cd] | 0.011 ± 0.002 [cde] | 0.041 ± 0.009 [bc] |
| | 3 | 0.040 ± 0.003 [ab] | 0.008 ± 0.002 [efg] | 0.048 ± 0.003 [b] |
| | 5 | 0.031 ± 0.000 [cd] | 0.009 ± 0.000 [def] | 0.040 ± 0.000 [bc] |
| | 9 | 0.020 ± 0.004 [ef] | 0.005 ± 0.001 [gh] | 0.025 ± 0.002 [de] |
| | 12 | 0.032 ± 0.003 [cd] | 0.009 ± 0.002 [def] | 0.041 ± 0.004 [bc] |

Means with different letters within a column are significantly different ($p < 0.05$).

## 3.5. Sensory Analysis

The sensory aesthetic characteristics of the product were assessed through an examination of the appearance of freshness, green color, thickness, and the numbers of brown spots. These factors play an important role in the acceptance of products by consumers [37]. The results of the survey of the sensory properties of the test lettuce during the 12 days of storage are shown in Table 4. It was found that the effect of interactions between packaging and storage time on all sensory properties of lettuce at the level of 1% was significant. The lettuce with the highest odor was identified as that in the control packaging on the first day, and the lowest odor was found in the lettuce in the 3% nano-packaging on the twelfth day of storage.

**Table 4.** The interaction effects of nano-packaging and storage time on the sensory properties of lettuce in MAP.

| Nano (%) | Time (Day) | Odor | Flavor | Appearance | Crispiness | Browning |
|---|---|---|---|---|---|---|
| control | 1 | 1.40 ± 0.52 [i] | 1.20 ± 0.42 [e] | 1.40 ± 0.52 [h] | 1.40 ± 0.52 [d] | 1.20 ± 0.42 [ef] |
| | 3 | 2.40 ± 0.52 [efg] | 1.20 ± 0.42 [e] | 1.80 ± 0.42 [gh] | 1.60 ± 0.52 [cd] | 1.60 ± 0.52 [de] |
| | 5 | 2.80 ± 0.42 [cde] | 1.60 ± 0.52 [de] | 2.20 ± 0.42 [fg] | 2.00 ± 0.00 [c] | 1.80 ± 0.42 [d] |
| | 9 | 3.00 ± 0.00 [bcd] | 3.20 ± 0.42 [a] | 4.60 ± 0.86 [bc] | 3.00 ± 0.00 [a] | 3.00 ± 0.00 [abc] |
| | 12 | 3.40 ± 0.52 [ab] | 3.20 ± 0.42 [a] | 5.00 ± 0.00 [a] | 3.00 ± 0.00 [a] | 3.20 ± 0.42 [ab] |
| 1 | 1 | 1.60 ± 0.52 [hi] | 1.20 ± 0.42 [e] | 1.40 ± 0.52 [h] | 1.40 ± 0.52 [cd] | 1.00 ± 0.00 [f] |
| | 3 | 2.20 ± 0.42 [fg] | 1.60 ± 0.52 [de] | 1.60 ± 0.52 [cd] | 1.60 ± 0.52 [de] | 1.40 ± 0.52 [def] |
| | 5 | 2.40 ± 0.42 [efg] | 1.80 ± 0.42 [cd] | 2.60 ± 0.52 [ef] | 1.80 ± 0.42 [cd] | 1.80 ± 0.42 [d] |
| | 9 | 2.60 ± 0.52 [def] | 1.80 ± 0.42 [cd] | 3.00 ± 0.47 [e] | 1.80 ± 0.42 [cd] | 1.80 ± 0.42 [d] |
| | 12 | 2.80 ± 0.42 [cde] | 3.00 ± 0.00 [a] | 4.20 ± 0.92 [cd] | 1.80 ± 0.42 [cd] | 1.80 ± 0.42 [d] |
| 3 | 1 | 1.60 ± 0.52 [hi] | 1.20 ± 0.42 [e] | 1.80 ± 0.42 [gh] | 1.60 ± 0.52 [cd] | 1.40 ± 0.52 [def] |
| | 3 | 2.00 ± 0.00 [gh] | 2.20 ± 0.42 [bc] | 3.80 ± 0.83 [d] | 2.40 ± 0.52 [b] | 2.60 ± 0.52 [c] |
| | 5 | 2.20 ± 0.52 [fg] | 2.20 ± 0.42 [bc] | 4.20 ± 0.79 [cd] | 2.40 ± 0.52 [b] | 2.80 ± 0.42 [bc] |
| | 9 | 3.20 ± 0.42 [bc] | 2.60 ± 0.52 [b] | 5.00 ± 0.00 [a] | 2.60 ± 0.52 [ab] | 3.20 ± 0.43 [ab] |
| | 12 | 3.80 ± 0.69 [a] | 3.20 ± 0.63 [a] | 5.80 ± 0.52 [a] | 3.00 ± 0.00 [a] | 3.40 ± 0.52 [a] |

Means with different letters within a column are significantly different ($p < 0.05$).

The lettuce samples had a suitable appearance in the control packaging and the 1% nano-packaging on the first day of storage. However, the 3% nano-packages showed the lowest quality of appearance of the lettuce on the twelfth day of storage. The changes in the appearance of the lettuce in the 1% nano-packaging were negligible (Table 4).

In terms of taste, all three types of packaging showed the best taste on the first day of storage. However, the weakest taste was found in the control packaging on the ninth day of storage and in the 1% and the 3% nano-packaging on the twelfth day of storage. The trend towards declining taste in the lettuce in the control packaging between the fifth day and the ninth day of storage was significantly greater than the same trend in the lettuce in the 1% nano-packaging on the first and twelfth days of storage (Table 4).

The highest degree of lettuce crispness was observed in the control packaging and the 1% nano-packaging. However, the lowest degree of crispness was found in the control packaging on the ninth and twelfth days of storage, and in the 3% nano-packaging on the twelfth day of storage. Changes in the 1% nano-packaging were not significant, whilst the trend for the other two packaging types was downward (Table 4).

In terms of the browning of the lettuce, the lowest browning rate was observed in the 1% nano-packaging on the first day of storage whilst the highest degree of browning was observed in the 3% nano-packaging on the twelfth day of storage. The browning rate in the 1% nano-packaging between the third day and the twelfth day of storage was not significant. Meanwhile, this trend in both the control packaging and the 3% nano-packaging was upward (Table 4).

*3.6. Microbiological Analysis*

3.6.1. *E. coli*

The results of the analysis of variance are shown in Table 5, indicating that there is a significant relationship between packaging $\times$ storage time and the growth of the bacterium. The bacterial growth decreased significantly in the control packaging. This can be related to the initial incompatibility of the microorganism with the new environment. However, after six days, the growth of *E. coli* in the control packaging increased compared to its growth in the other two types of packaging. Although all three films had a negative effect on the growth of *E. coli*, their reduction rates were different. In the 1% nano-packaging, the initial population of microorganisms was 2.76 Log cfu/g by the fourth day, a figure that remained stable until the sixth day. However, in the following days, the growth of *E. coli* increased significantly. In the 3% nano-packaging, the initial population of *E. coli* was 2.98 Log cfu/g, and this remained stable until the eighth day of storage. These results indicated that the 3% nano-packaging had a higher inhibitory effect on the growth of *E. coli*.

**Table 5.** The interaction effects of nano-packaging and storage time on *E. coli* growth (Log CFU/g) in lettuce in MAP.

| Storage Time (Day) | Control | Nano 1% | Nano 3% |
|---|---|---|---|
| 0 | 9.35 ± 0.08 [a] | 9.15 ± 0.22 [a] | 9.18 ± 0.33 [a] |
| 2 | 7.33 ± 0.03 [de] | 8.10 ± 0.01 [bc] | 8.32 ± 0.30 [b] |
| 4 | 6.42 ± 0.22 [gh] | 6.63 ± 0.40 [fgh] | 7.60 ± 0.78 [cd] |
| 6 | 6.26 ± 0.23 [h] | 6.39 ± 0.22 [gh] | 6.59 ± 0.59 [fgh] |
| 8 | 7.02 ± 0.33 [ef] | 6.86 ± 0.03 [efg] | 6.20 ± 0.10 [h] |
| 10 | 7.57 ± 0.16 [cd] | 8.05 ± 0.05 [bc] | 7.25 ± 0.05 [de] |

Means with different letters within a column are significantly different ($p < 0.05$).

3.6.2. Mold and Yeast

The results of analysis of variance indicated that the interaction effects (packing $\times$ storage time) on yeast growth were statistically significant at the 1% level. No growth of yeast was reported in the packages on the first day of storage. Additionally, in the 1% nano-packaging, no growth was observed until the fifth day of storage, whilst the

highest yeast growth was observed in the 3% nano-packaging on the fifth day of storage. The lowest growth after 12 days of storage was in the 1% and the 3% nano-packaging (Table 6). In the control packaging, there were no significant changes in the yeast population between the fifth and tenth days of storage. The growth of yeast in the 3% nano-packaging decreased over the test period. No mold growth was observed in the packages during the 15 days of storage. In terms of yeast, the highest inhibitory effect was achieved with the 1% nano-packaging, with no growth until the fifth day of storage. In the other two packaging types, the growth of yeast started from the fifth day of storage and continued until the tenth day. After the tenth day, owing to a decrease in the percentage of oxygen required for growth in the environment, the yeast growth decreased.

**Table 6.** The interaction effects of nano-packaging and storage time on yeast growth (Log CFU/g) in lettuce in MAP.

| Time (Day) | Control | Nano 1% | Nano 3% |
|---|---|---|---|
| 0 | 0.00 ± 0.00 [e] | 0.00 ± 0.00 [e] | 0.00 ± 0.00 [e] |
| 5 | 3.51 ± 0.10 [b] | 0.00 ± 0.00 [e] | 4.54 ± 0.36 [a] |
| 10 | 3.52 ± 0.03 [b] | 3.18 ± 0.18 [c] | 3.48 ± 0.10 [b] |
| 15 | 3.18 ± 0.08 [c] | 2.04 ± 0.04 [d] | 2.02 ± 0.20 [d] |

Means with different letters within a column are significantly different ($p < 0.05$).

In a study on asparagus packaged in nano-packages, it was found that the growth of mold and yeast increased in packaging after 15 days. Rinsing with water can reduce the yeast and fungal growth up to 0.87 Log cfu/g [35]. Additionally, using sodium hypochlorite (200 ppm), the population of yeast and fungal growth decreased significantly to 2.75 Log cfu/g [39]. As the evidence and the experiments in this study show, no growth of mold was observed in the 1% nanofilm packages on the fifth day, but for longer storage times the 1 and the 3% packages were similar. Fresh products are living, respiring tissues. Thus, oxygen is consumed to produce carbon dioxide that is of itself anti-fungal and anti-bacterial; the significant reduction in the growth of mold and yeast in the package containing nanoparticles compared to the control samples can be attributed to this accumulation of carbon dioxide [26].

## 4. Conclusions

In this study, the physicochemical and antimicrobial characteristics of polypropylene films containing various percentages of bentonite nanoparticles were investigated. Our results demonstrated that the application of a low percentage of bentonite nanoparticles could significantly enhance the microbial quality of iceberg lettuce samples via the reduction of *E. coli*, mold, and yeast counts. Additionally, the presence of nano-bentonite in polypropylene packaging could significantly enhance the shelf life of iceberg lettuce, as evidenced by the physicochemical characteristics of samples stored under refrigerated conditions. Consequently, based on the results of the current study, further exploration of the application of a wider range of percentages of nano-bentonite used in conjunction with MAP packaging is suggested.

**Author Contributions:** Conceptualization, Z.F. and N.Z.; methodology, Z.F.; software, Z.F.; validation, Z.F., N.Z., G.G. and E.K.; formal analysis, Z.F.; investigation, Z.F. and S.P.; resources, S.P.; data curation, Z.F. and G.G.; writing—original draft preparation, Z.F. and N.Z.; writing—review and editing, G.G. and N.T; visualization, N.Z.; supervision, N.Z.; project administration, N.Z.; funding acquisition, N.T. All authors have read and agreed to the published version of the manuscript.

**Funding:** This research received no external funding.

**Institutional Review Board Statement:** Not applicable.

**Informed Consent Statement:** Not applicable.



**Data Availability Statement:** Not applicable.

**Conflicts of Interest:** The authors declare no conflict of interest.

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
