# Peer review of "Effects of Nano-Bentonite Polypropylene Nanocomposite Films and Modified Atmosphere Packaging on the Shelf Life of Fresh-Cut Iceberg Lettuce"

_coatings, doi:10.3390/coatings13020349_

Round 1

Reviewer 1 Report

The study deals with the use of nano-packaging materials for the extension of the shelf life of fresh-cut iceberg lettuce. Although the study is relevant and has potential, there is a need for further improvement, especially on the methodology and the presentation of the results. 

Specific comments are given below:

1. No specific information is given on how the lettuce was cut into pieces.

2. No information about the microbiological quality of the raw lettuce before cutting. The zero point is when the lettuce has been packed, however, it is important to know the microbial counts of the lettuce prior cutting.

3. The concentration of the ascorbic acid is not given in the materials and methods. 

4. Three replications are mentioned in the methodology. These are technical replicates? Biological replicates? Was the experiment performed with different batches of lettuce? At different seasons?

5. Why exudates were measured? Why this info is important? This needs to be included.

6. Section 2.4.4. Total soluble solids or dry matter? These are different characteristics. 

7. Section 2.4.7. Texture is a very general term. Maybe firmness could better describe the analysis? Or another term to make it more specific?   

8. Line 172: How was the panel trained?

9. The scale used for sensory analysis was 1-5 or 1-9? There is confusion in this part. Also the whole sensory evaluation method needs to be rewritten. It is not described clearly. 

10. Line 180: Proposed by? The names of the authors are missing. 

11. Line 186: Technical replications or biological replications? Similar issue as before. 

12. What is TBC medium? The full name of the medium is needed.

13. In section 3.5.1. How is the increase of E. coli counts explained on day 10? One would expect that since pH is decreasing,  E. coli would be eliminated slowly slowly. Check again your data.

14. Why yeasts were checked for up to 15 days, while E. coli for up to 10 days? And the product for up to 12 days? 

Author Response

Many thanks for your suggestions and observations on how to improve our paper. Please see the attached file for our specific responses. 

Reviewer 2 Report

Dear, the paper entitled "Effects of nano-bentonite low density polyethylene

nanocomposite films and modified atmosphere packaging on the shelf life of fresh cut iceberg lettuce" needs some adjustments to be approved.

I would like to see what these packages look like. There are no photos in the document.

The authors present several tables and graphs comparing the 2 formulations with different amounts of bentonite, but there is no photographic accompaniment, so that these alterations can be observed.

Microbiological experiments are confusing. The authors place a number of microorganisms and then follow them over time. Values fluctuate a lot, decrease over time, then increase. I don't think that with the data in the table we can say that bentonite is effective, because in some moments, the control shows smaller, significant values.

The count of molds and yeasts, on the other hand, is not done by inoculating the food, but by observing those in the environment. When analyzing table 6, the results of 1 and 3% are very discrepant for 5 days, and I see difficulties in concluding on these data.

There are very few cited bibliographical references, and they are very old. I suggest a good update.

Author Response

(The authors gave the same response as above.)

Reviewer 3 Report

The authors used nano bentonite as filler for low density polyethylene and found that the resulted composite film effective as packaging for iceberg lettuce. I suggest acceptance after minor revision on the following aspects

1.        The Introduction failed to introduce the current work smoothly. Research status of conventional polymer with bentonite nanocomposites as food packaging should be given; How does the current work contribute to the field?

2.        Preparation of the PE-bentonite nanocomposite film seems missing.

3.        Figure 1 contains two Figures which should be labeled and discussed accordingly.  Current discussion (on pH) doesn’t harmonize with the figures(showing oxygen barrier)

Author Response

(The authors gave the same response as above.)

Reviewer 4 Report

This manuscript describes the effects of nano-bentonite low density polyethylene nanocomposite films and modified atmosphere packaging on the shelf life of fresh cut iceberg lettuce. It was very interesting. However, some minor revision should be done to meet the requirement for publishment.

1. Abstract: The greatest growth of E. coli was observed in control films which is 27 demonstrates the effectiveness of the application of bentonite nanoparticle fillers. 27 demonstrates should be used to compare with other groups or published references.

2. The authors mentioned that nano-packaging had a higher ability in maintaining dry matter of samples compared to control packaging. Please add the reason in the discussion.

3. The browning rate in the 1% nano-packaging between the third day and the twelfth day of storage was not significant. Meanwhile, this trend in both the control packaging and 3% nano-packaging was upward. Why select the third day and the twelfth day?

4. 1% nano-packaging shows a high ability to maintain the quality of lettuce texture until the fifth day of storage[31]. It’s confused that this sentence is the conclusion or discussion from a reference of 31.

Author Response

(The authors gave the same response as above.)

Round 2

Reviewer 2 Report

I redid the reading of the document and noticed that some points are still not clarified. I would like to see photos of all groups. Only the control was sent as an attachment. Bibliographical references have not been changed.

Author Response

We have updated format of the manuscript to include the figures and pictures as requested. The bibliographical references have been revised in the MS.

Best regards.
